# Quasi-Maximum Likelihood Estimation for Long Memory Stock Transaction Data—Under Conditional Heteroskedasticity Framework

**A. M. M. Shahiduzzaman Quoreshi** [1,*] **, Reaz Uddin** [1] **and Naushad Mamode Khan** [2]

[1] Department of Industrial Economics, Blekinge Institute of Technology, SE-371 79 Karlskrona, Sweden; reaz_uddin@yahoo.com

[2] Department of Economics and Statistics, Faculty of Social Sciences and Humanities, University of Mauritius, Reduit 80837, Mauritius; n.mamodekhan@uom.ac.mu

* Correspondence: shahiduzzaman.quoreshi@bth.se; Tel.: +46-734-223-619

**Abstract:** This paper introduces Quasi-Maximum Likelihood Estimation for Long Memory Stock Transaction Data of unknown underlying distribution. The moments with conditional heteroscedasticity have been discussed. In a Monte Carlo experiment, it was found that the QML estimator performs as well as CLS and FGLS in terms of eliminating serial correlations, but the estimator can be sensitive to start value. Hence, two-stage QML has been suggested. In empirical estimation on two stock transaction data for Ericsson and AstraZeneca, the 2SQML turns out relatively more efficient than CLS and FGLS. The empirical results suggest that both of the series have long memory properties that imply that the impact of macroeconomic news or rumors in one point of time has a persistence impact on future transactions.

**Keywords:** count data; estimation; finance; high frequency; intraday; time series

**JEL Classification:** C13; C22; C25; C51; G12; G14

## 1. Introduction

Classical economic theory of price determination is a function of demand and supply. For example, in the Walrasian auctioneer approach, demands and supplies of a good are aggregated to find a market-clearing price. However, availability of high frequency data enables studying market mechanisms or market microstructures. These studies depart from demand–supply function of price determination under classical economics and takes into account other factors considered to be influential in price determination. For instance, Working (1953) in matching demand and supply curves in equilibrium expands focus on the underlying trading mechanism. Demsetz (1968) investigates effect of transactions costs in price determination in securities market and incorporates the influence of time factor of demand and supply in analyzing formation of market prices. Studies of market microstructure concern, among others, the impact of transactions, bid–ask spreads (the difference between bid and ask prices), volume of and time between transactions (duration) on price formation. The studies also investigate trading behavior of actors in response to news, rumors, etc. In the securities market, a transaction (or a trade) is completed by a buyer and a seller agreeing to exchange a specific volume of stocks at certain price. These transactions and time elapsed between each transaction are correlated. Fewer trades take place with increasing time lapse between successive transactions in a given time interval. Therefore, trading intensity and duration can be considered to be inversely related. During the past decades research on market microstructure to understand pricing processes has been centered on the trading intensity and durations. Diamond and Verrecchia (1987) illustrate implications of bad

news in low trading intensity. Easley and O'hara (1992) differ on level of implication demonstrating that a low trading intensity implies no news. Besides, Engle (2000) finds that durations are associated with price volatilities. Moreover, the stock transactions data are counts for a fixed interval of time and Quoreshi (2014) explains that the time series of the data may also have long memory property.

The long memory phenomenon in time series has been first considered by Hurst (1951, 1956). In these studies, he explains the long term storage requirements of the Nile River. He shows that the cumulated water flows in a year depends not only on the water flows in recent years, but also on water flows in years much prior to the present year. Mandelbrot and van Ness (1968) explain and advance Hurst's studies by employing fractional Brownian motion. In analogy with Mandelbrot and van Ness (1968); Granger (1980); Granger and Joyeux (1980) and Hosking (1981) developed Autoregressive Fractionally Integrated Moving Average (ARFIMA) models to account for the long memory in time series data. Granger and Ding (1996) pointed out that a number of other processes can also have the long memory property. An empirical study regarding the usefulness of ARFIMA models is conducted by Bhardwaj and Swanson (2005), who find strong evidence in favor of ARFIMA in absolute, squared, and log-squared stock index returns.

A time series of count data describes a non-negative sequence of count observations, which is integer-valued and observed at equidistant intervals of time. The literature on different techniques to model, estimate and exploit such data is ever growing. Jacobs and Lewis (1978a, 1978b, 1983) introduced time dependence and developed discrete ARMA (DARMA) models. An important difference between the continuous variable ARMA model and its corresponding Integer-valued ARMA (INARMA) version is that the latter contains parameters that are interpreted as probabilities and then take on values in narrower intervals than do the parameters of the ARMA model (e.g., McKenzie 1986; Al-Osh and Alzaid 1987; Al-Osh and Alzaid 1988). Brännäs and Quoreshi (2010) advance integer-valued moving average model to model the number of transactions in intraday data of stocks. Quoreshi (2006, 2008, 2014, 2017) advances further the INMA model into bivariate, multivariate and long memory (INARFIMA) framework. These papers consider Conditional Least Square (CLS), Feasible Generalized Least Square (FGLS), and Generalized Methods of Moments (GMM). A large number of studies have considered the modeling of bivariate or multivariate count data assuming an underlying Poisson distribution (e.g., Gourieroux et al. 1984). Heinen and Rengifo (2003) introduced multivariate time series count data models based on the Poisson and the double Poisson distribution. Other extensions to traditional count data regression models are considered by, e.g., Brännäs and Brännäs (2004) and Rydberg and Shephard (1999). None of these papers consider maximum likelihood estimation since full density function of underlying distribution is unknown. Sunecher et al. (2018) recently introduce a first-order bivariate integer-valued moving average process (BINMA(1)) where they propose a generalized quasi-likelihood (GQL) method of estimation. Ristic et al. (2018) introduces a new bivariate integer-valued moving average of the first order (BINMA(1)) process with independent Negative Binomial (NB) innovations under nonstationary moment conditions. They also employed a generalized quasi-likelihood (GQL) method of estimation.

In this paper, we propose a quasi-maximum likelihood (QML) estimator for nonstationary integer-valued long memory model for unknown underlying distribution and compare this estimator with CLS and FGLS that have performed better than GMM in the previous studies. We employ the QML estimator on stock transactions data for Ericsson and AstraZeneca. Both stock series demonstrate long memory. Empirically it is also found that the QML estimator is more efficient compared to the other estimators.

The paper is organized as follows. The INARFIMA $(0, d, 0)$ model is presented in Section 2. The conditional and unconditional moment properties of the INARFIMA $(0, d, 0)$ models are obtained. The estimation procedures, CLS and FGLS for unknown parameters are discussed and the QML estimator is proposed in Section 3. The results from a Monte Carlo experiment are presented in Section 4. A detailed description of the empirical data is given in Section 5. The empirical results for the stock series are presented in Section 6, and the concluding comments are included in the final section.

## 2. The Model

Quoreshi (2014) proposes the INARFIMA $(p, d, q)$ long memory properties of stock transaction data. The INMA ($\infty$) representation of INARFIMA model, which is INARFIMA $(0, d, 0)$, can be written as

$$y_t = u_t + d_1 \cdot u_{t-1} + d_2 \cdot u_{t-2} + d_3 \cdot u_{t-3} \ldots$$

or

$$y_t = (1 + L^{\bullet})^{-d} u_t \tag{1}$$

where $L$ is a lag operator and the notation $L^{\bullet} = (\bullet L)^i$ for $i > 0$. Here we assume that here is no cross-lag dependence among $u_t$. Note that $y_t$ has long memory in a sense that the variables have slow decaying autocorrelation functions and the parameters $d_j = \Gamma(j + d_i) / [\Gamma(j + 1)\Gamma(d_i)]$, $j = 0, 1, 2, \ldots$ where $d_0 = 1$. Note that $d_j$ are considered thinning probabilities and hence $d_j \in [0, 1]$. The macroeconomic news or rumors are assumed to be captured by $\{u_t\}$ and filtered by $\{d_j\}$ through the system. The binomial thinning operator is used to account for the integer-valued property of count data. This operator can be written as

$$\alpha \cdot u = \sum_{j}^{u} z_j \tag{2}$$

with $\{z_j\}_{j=1}^{u}$ an iid sequence of 0–1 random variables, such that $\Pr(z_j = 1) = \alpha = 1 - \Pr(z_j = 0)$. Conditionally on the integer-valued $u$, $\alpha \cdot u$ is binomially distributed with $E(\alpha \cdot u | u) = \alpha u$ and $V(\alpha \cdot u | u) = \alpha(1 - \alpha)u$. Unconditionally it holds that $E(\alpha \cdot u) = \alpha \lambda$ and $V(\lambda \cdot u) = \alpha^2 \sigma^2 + \alpha(1 - \alpha)\lambda$, where $E(u) = \lambda$ and $V(u) = \sigma^2$. Obviously, $\alpha \cdot u \in \{0, 1, \ldots, u\}$. Many authors accentuate exact distributional results for $y_t$, while Brännäs and Quoreshi (2010) stress only the first- and second-order moment conditions. In analogy with Brännäs and Quoreshi (2010), we construct the Quasi-Maximum Likelihood Estimation based on the first and second-order moment condition.

Assuming independence between and within the thinning operations and $\{u_t\}$ an iid sequence with mean $\lambda$ and variance $\sigma^2 = v\lambda$ where $v > 0$, the unconditional first and second-order moments can be given as

$$E(y_t) = \lambda \left( 1 + \sum_{i=1}^{\infty} d_i \right) \tag{3}$$

$$V(y_t) = \lambda \left[ \left( v + \sum_{i=1}^{\infty} d_i \right) + (v - 1) \left( 1 + \sum_{i}^{\infty} d_i^2 \right) \right] \tag{4}$$

$$\gamma_k = \sigma^2 \left( d_k + \sum_{i=1}^{\infty} d_i d_{k+1} \right), \qquad k \geq 1 \tag{5}$$

where $\gamma_k$ denotes the autocovariance function at lag $k$. It is obvious from (3)–(5) that the mean, variance, and autocovariance take only positive values since $\lambda$, $\sigma^2$, and $d_i$ are all positive and that $\sum_{i=1}^{\infty} d_i < \infty$ is required for $\{y_t\}$ to be a stationary process. Note also that the variance may be larger than the mean (overdispersion), smaller than the mean (underdispersion), or equal to the mean (equidispersion) depending on whether $v > 1$, $v \in (0, 1)$ or $v = 1$, respectively. When lag length $q$ is finite, summing to infinite is replaced to summing to $q$. The conditional mean, variance, and covariance for the INARFIMA $(0, d, 0)$ are in an analogous way:

$$E(y_t | Y_{t-1}) = E_{t-1} = \lambda \left( 1 + \sum_{i=1}^{\infty} d_i u_{t-i} \right) \tag{6}$$

$$V(y_t | Y_{t-1}) = V_{t-1} = \lambda v + \sum_{i=1}^{\infty} d_i (1 - d_i) u_{t-i} \tag{7}$$

where $Y_{t-1}$ is the information set available at time $t - 1$. The conditional mean and variance vary with $u_{t-i}$. Since the conditional variance varies with $u_{t-i}$, there is a conditional heteroscedasticity or nonstationary property of moving average type that Brännäs and Hall (2001) called MACH(q). The effect of $u_{t-i}$ on the mean is greater than on the variance. Note also, that like the unconditional variance, the conditional variance could be overdispersed, underdispersed, or equidispersed depending on whether $v > 1 + \sum_i^\infty \frac{d}{\lambda}$, $v < (1 + \sum_i^\infty \frac{d_i}{\lambda})$ or $v_i = 1 + \sum_i^\infty \frac{d_i}{\lambda}$, respectively.

## 3. Estimation

If we do not assume a full density function, we may estimate the Quasi Maximum Likelihood (QML) Estimator as discussed by Weiss (1986) instead of Maximum Likelihood (ML) Estimator. Conditional Least Square (CLS), Feasible Generalized Least Square (FGLS) are Generalized Methods of Moments discussed in INMA model (Brännäs and Quoreshi 2010). In the previous studies, FGLS is the best estimator among the three in terms of eliminating serial correlation (Brännäs and Quoreshi 2010). The CLS is second, which is almost as good as FGLS. Here, we only construct the QML and FGLS for INARFIMA $(0, d, 0)$ model and compare the results with CLS.

The Conditional least square (CLS) estimator for INARFIMA $(0, d, 0)$ representation model has the following residual.

$$e_t = y_t - E_{t-1} = y_t - \lambda \left( 1 + \sum_{i=1}^{\infty} d_i u_{t-i} \right) \tag{8}$$

The criterion function $S_{CLS} = \sum_{i=m+1}^{T} e_t^2$ is minimized with respect to unknown parameters, i.e., $\psi = (\lambda$ and $\delta')$ where $\delta'$ is vector of parameters with elements $d_i$. The $E_{t-1}$ is the conditional expected value of $y_t$ defined in Equation (6). Using a finite maximum lag $m$ in (8) instead of infinite lags may cause biasing effects. Due to omitted variables, i.e., $u_{t-m-1}, \ldots, u_{t-\infty}$, we may expect a positive bias on the parameters $\lambda$ and $\delta'$ (Brännäs and Quoreshi 2010). These moment conditions correspond to the normal equations of the CLS estimator that focuses on the unknown parameters of the conditional mean function. Alternatively and equivalently, the properties $E(e_t) = 0$ and $E(e_t e_{t-j}) = 0$, $j \geq 1$ could be used. The FGLS estimator minimizes

$$S_{FGLS} = \sum_{t=m+1}^{T} e_t \hat{V}^{-1} \tag{9}$$

with $\hat{V}^{-1}$ as given. The variance of error from CLS estimates may be used for approximation of $\hat{V}^{-1}$ in equation (9). Alternatively, $\hat{V}^{-1}$ can be estimated as specified in (7) by employing estimates from CLS. The covariance matrix estimators for CLS and FGLS are

$$Cov\left(\hat{\psi}_{CLS}\right) = \left( \sum_{t=m+1}^{T} \frac{\partial e_t}{\partial \psi} \frac{\partial e_t}{\partial \psi'} \right)^{-1}$$

$$Cov\left(\hat{\psi}_{FGLS}\right) = \left( \sum_{t=m+1}^{T} \hat{V}^{-1} \frac{\partial e_t}{\partial \psi} \frac{\partial e_t}{\partial \psi'} \right)^{-1}$$

The QML estimator for INARFIMA $(0, d, 0)$ representation model has the same residual as in Equation (9), and we propose the following criterion function to maximize

$$f(y_1, y_2 \ldots, y_T | Y_{t-1}, \lambda \text{ and } \delta') = \prod_{t=1}^{T} f(y_t | Y_{t-1}, \psi_i) = \left( \frac{1}{2\pi V_{t-1}} \right)^{T/2} exp\left( -\frac{\sum_{i=m+1}^{T} e_t^2}{2V_{t-1}} \right) \tag{10}$$

where $\psi_i = (\lambda$ and $\delta')$ and $V_{t-1}$ is as in Equation (7). This specification may be motivated by the central limit theorem since $y_t$ are counts but not the $E_{t-1}$. According to the central limit theorem, the standardized expected value of a sample is normally distributed with mean zero and variance one if the sample size is large enough. A relevant empirical study of distribution properties on high-frequency intraday transaction prices are conducted by Andersen et al. (2001). Taking the logarithm of Equation (10), we may simply use the criterion function and minimize the function as

$$Lnf\left(y_1,\ y_2\ldots,y_T\middle|Y_{t-1},\lambda,\delta'\text{and}\hat{V}_{t-1}\right) = -\frac{T}{2}\ln\left(\hat{V}_{t-1}\right) - \ln(2\pi) - \left(\frac{\sum_{t=m+1}^{T}e_t^2}{2\hat{V}_{t-1}}\right) \quad (11)$$

where $\hat{V}_{t-1}$ is an estimate for $V_{t-1}$ that is to be estimated. Since $T$, 2, and $\pi$ are constants, we can equivalently minimize the following criterion function

$$Lnf\left(y_1,\ y_2\ldots,y_T\middle|Y_{t-1},\lambda,\delta'\text{and}\hat{V}_{t-1}\right) = -\ln\left(\hat{V}_{t-1}\right) - \left(\frac{\sum_{t=m+1}^{T}e_t^2}{\hat{V}_{t-1}}\right). \quad (12)$$

Note that the $\hat{V}_{t-1}$ is to be estimated at the same time as the other parameters. If the estimation is sensitive to the start value of $\hat{V}_{t-1}$, we can obviously estimate CLS at the first stage and calculate the $\hat{V}_{t-1}$ which can be used as the start value for QML. We call this estimation procedure Two-Stage Quasi Maximum Likelihood (2SQML) Estimation. The covariance matrix estimators for QML and 2SQML are

$$Cov\left(\hat{\psi}_{QML}\right) = \left(\sum_{t=m+1}^{T}\hat{V}^{-1}\frac{\partial e_t}{\partial\psi}\frac{\partial e_t}{\partial\psi'}\right)^{-1}$$

$$Cov\left(\hat{\psi}_{2SQML}\right) = \left(\sum_{t=m+1}^{T}\hat{V}_{CLS}^{-1}\frac{\partial e_t}{\partial\psi}\frac{\partial e_t}{\partial\psi'}\right)^{-1}.$$

Note that we call an estimator relatively efficient if we have a smaller standard deviation (error) for the parameters of interest compared to the other estimator. The covariance matrices mentioned above calculate the standard error for the estimates in respective estimator. The criteria that are used to choose best model fitting are, e.g., Adjusted-$R^2$, mean square error (MSE), Akaike Information Criteria (AIC), and Schwarz Bayesian Information Criterion (SBIC). The Adjusted-$R^2$ explains the degree of variation in the dependent variable explained by the independent variables, while MSE explains to what extend the regression models are unable to explain the variation of independent variable. Hence, this implies that the higher Adjusted-$R^2$ the smaller MSE. The $S_{CLS}$ is the MSE for Equation (8), while the corresponding Adjusted-$R^2$ is $(1 - S_{CLS}/\text{var}(y_t))$, where $\text{var}(y_t)$ is the variance of the independent variable. The ACI and SBIC are similar to MSE. Instead of MSE, the AIC and SBIC use a function that incorporate the value from the likelihood function used in estimation, as in Equation (11). In the first step, these criteria may be used for model selection.

In time series, Ljung–Box statistics and Box–Pierce test are widely used to check the serial correlations between the residuals. These criteria can be used to evaluate estimators given a model. The null hypothesis is that the residuals $e_t$ are independently distributed while the alternative hypothesis is that the residuals are not independently distributed; there are serial correlations between the residuals. The Ljung–Box statistics for residuals can be written

$$Q_{LB} = n(n+2)\sum_{k=1}^{h}\frac{\hat{\rho}_k^2}{n-k}$$

where $n$ is the number of observations and $k$ is the number of lags used for estimation (Ljung and Box 1978). The $\hat{\rho}_k$ is the autocorrelations of residuals at lag $k$. The null hypothesis is rejected if

$$Q_{LB} > \chi^2_{1-\alpha,h}$$

where $\chi^2_{1-\alpha,h}$ is chi-squared distribution with significance level $\alpha$ and degree of freedom $h$. Similarly, the Box–Pierce test can be written as

$$Q_{BP} = n \sum_{k=1}^{h} \hat{\rho}_k^2$$

where $n$, $k$, are $\hat{\rho}_k$ as described above (Box and Pierce 1970). The difference between these two statistics emerges from the term $(n+2)$ and $(n-k)$ used in Ljung–Box statistics and $Q_{LB} > Q_{BP}$ and hence $Q_{LB}$ is more restrictive to reject the null hypothesis.

## 4. Monte Carlo Experiment

Quoreshi (2014) studies, in a brief Monte Carlo experiment, the bias, MSE, Ljung–Box statistics (LB), AIC, and SBIC properties of the CLS estimators for finite-lag specifications; data is generated according to INARFIMA $(0, d, 0)$. Here, we study employing QML, 2SQML, and FGLS estimators and compare the results with CLS. Initially, we found that QML is highly sensitive to start values and produces large biased estimates. Hence, we only focus the result on 2SQML, FGLS, and CLS. This study generates data from Poisson distribution to study the bias properties under 2SQML and FGLS estimation procedure. Smith et al. (1996) and Quoreshi (2014) studied the bias and misspecification in ARFIMA and INARFIMA models, respectively. Drost et al. (2009) investigate finite sample behavior of semiparametric integer-valued AR(p) models, while Brännäs and Quoreshi (2010) study finite lag misspecification when the data is generated according to an infinite-lag INMA model. In this Monte Carlo experiment we study the bias, MSE, Ljung–Box statistics, AIC, and SBIC properties of the FGLS and 2SQML estimators for finite-lag specifications when data is generated according to INARFIMA $(0, d, 0)$. The data generating process is as in (1), with $d_j = \Gamma(j+d)/[\Gamma(j+1)\Gamma(d)]$, $j = 0, 1, 2, \ldots$ where $d_0 = 1$ and where $u_t$ is drawn according to Equation (2). The values for $d$, $d = 0.1$, 0.25 and 0.4 are used and lag length $m = 70$ is chosen. The $u_t$ sequence is generated as Poisson with parameter $\lambda = 5$ so that $\lambda = \sigma^2$ in the conditional variance equation in (7). Six time series with length $T = 2000$ and $T = 10,000$ are generated. The first 500 observations are discarded to avoid the start-up effect. The results for the Monte Carlo experiment are given in Tables 1–3. In Table 1, we see that the estimates for $d$ decreases as lag length (M) increases from 10 to 70 and approaches to the theoretical value $d = 0.1$. This implies that the biasness of the estimates decreases from 0.071 to 0.004 for $T = 2000$ as the lag length approaches the theoretical lag length 70 (see $\hat{d}$ corresponding to M10–M70 in Table 1). The parameter estimated with lag length 90 (M90) is $d < 0.1$, which implies negative biasness. Like Brännäs and Quoreshi (2010), we conclude that we may expect a positive biasing effect on the parameters due to omitted variables. The results for MSE, AIC, and SBIC for all the three estimators are same up to three decimals (Table 1). Depending on the size of $d$, the standard AIC and SBIC may need to be corrected. The result indicates that 2SQML, FGLS and CLS estimators perform equally well in terms of eliminating serial correlation (see $Q_{LB100}$ and $Q_{LB100}$ in Tables 1–3). However, standard error (see s.e. for M70 in Tables 1–3) for $d$ varies slightly between the estimator. Taking this into account, we may conclude that CLS performs best while 2SQML estimator performs better than FGLS. It is to be noted that both QML and FGLS are sensitive to start values. In this case 2SQML should be used.

**Table 1.** The properties of the Conditional Least Square (CLS), Feasible Generalized Least Square (FGLS), and Two-Stage Quasi Maximum Likelihood (2SQML) estimators for finite-lag specifications, when data is generated according to Integer-valued Autoregressive Fractionally Integrated Moving Average (INARFIMA) (0,*d*,0) models with $d = 0.1$ and lag (M) = 70.

| Lag | Parameters | $T$ = 2000 and $d$ = 0.1 | | | $T$ = 10,000 and $d$ = 0.1 | | |
|---|---|---|---|---|---|---|---|
| | | **CLS** | **FGLS** | **ML** | **CLS** | **FGLS** | **ML** |
| M10 | $\hat{d}$ | 0.171 | 0.171 | 0.171 | 0.167 | 0.167 | 0.167 |
| | (s.e.) | 0.001 | 0.003 | 0.003 | 0.000 | 0.001 | 0.001 |
| | BIAS | 0.071 | 0.071 | 0.071 | 0.067 | 0.067 | 0.067 |
| | MSE | 8.286 | 8.286 | 8.286 | 8.117 | 8.117 | 8.117 |
| | $Q_{LB100}$ | 135.447 | 135.447 | 135.447 | 207.628 | 207.628 | 207.628 |
| | $Q_{LB200}$ | 210.680 | 210.680 | 210.680 | 302.137 | 302.137 | 302.137 |
| | AIC | 4239.785 | 4239.785 | 4239.785 | 20,943.299 | 20,943.299 | 20,943.299 |
| | SBIC | 4312.395 | 4312.395 | 4312.395 | 21,033.613 | 21,033.613 | 21,033.613 |
| M30 | $\hat{d}$ | 0.126 | 0.126 | 0.126 | 0.123 | 0.123 | 0.123 |
| | (s.e.) | 0.001 | 0.002 | 0.002 | 0.000 | 0.001 | 0.001 |
| | BIAS | 0.026 | 0.026 | 0.026 | 0.023 | 0.023 | 0.023 |
| | MSE | 8.226 | 8.226 | 8.226 | 8.034 | 8.034 | 8.034 |
| | $Q_{LB100}$ | 127.753 | 127.753 | 127.753 | 129.387 | 129.387 | 129.387 |
| | $Q_{LB200}$ | 200.223 | 200.223 | 200.223 | 220.119 | 220.119 | 220.119 |
| | AIC | 4246.408 | 4246.408 | 4246.408 | 20,867.821 | 20,867.821 | 20,867.821 |
| | SBIC | 4451.036 | 4451.036 | 4451.036 | 21,122.341 | 21,122.341 | 21,122.341 |
| M50 | $\hat{d}$ | 0.112 | 0.112 | 0.112 | 0.109 | 0.109 | 0.109 |
| | (s.e.) | 0.001 | 0.002 | 0.002 | 0.000 | 0.001 | 0.001 |
| | BIAS | 0.012 | 0.012 | 0.012 | 0.009 | 0.009 | 0.009 |
| | MSE | 8.188 | 8.188 | 8.188 | 8.018 | 8.018 | 8.018 |
| | $Q_{LB100}$ | 124.244 | 124.244 | 124.244 | 110.770 | 110.770 | 110.770 |
| | $Q_{LB200}$ | 197.235 | 197.235 | 197.235 | 199.517 | 199.517 | 199.517 |
| | AIC | 4257.168 | 4257.168 | 4257.168 | 20,868.726 | 20,868.726 | 20,868.726 |
| | SBIC | 4593.814 | 4593.814 | 4593.814 | 21,287.453 | 21,287.453 | 21,287.453 |
| M70 | $\hat{d}$ | 0.104 | 0.104 | 0.104 | 0.101 | 0.101 | 0.101 |
| | (s.e.) | 0.001 | 0.002 | 0.002 | 0.000 | 0.001 | 0.001 |
| | BIAS | 0.004 | 0.004 | 0.004 | 0.001 | 0.001 | 0.001 |
| | MSE | 8.190 | 8.190 | 8.190 | 8.016 | 8.016 | 8.016 |
| | $Q_{LB100}$ | 122.988 | 122.988 | 122.988 | 104.384 | 104.384 | 104.384 |
| | $Q_{LB200}$ | 195.961 | 195.961 | 195.961 | 193.215 | 193.215 | 193.215 |
| | AIC | 4277.888 | 4277.888 | 4277.888 | 20,886.425 | 20,886.425 | 20,886.425 |
| | SBIC | 4746.552 | 4746.552 | 4746.552 | 21,469.359 | 21,469.359 | 21,469.359 |
| M90 | $\hat{d}$ | 0.099 | 0.099 | 0.099 | 0.096 | 0.096 | 0.096 |
| | (s.e.) | 0.001 | 0.002 | 0.002 | 0.000 | 0.001 | 0.001 |
| | BIAS | −0.001 | −0.001 | −0.001 | −0.004 | −0.004 | −0.004 |
| | MSE | 8.168 | 8.168 | 8.168 | 8.021 | 8.021 | 8.021 |
| | $Q_{LB100}$ | 121.982 | 121.982 | 121.982 | 99.539 | 99.539 | 99.539 |
| | $Q_{LB200}$ | 192.315 | 192.315 | 192.315 | 188.162 | 188.162 | 188.162 |
| | AIC | 4292.417 | 4292.417 | 4292.417 | 20,911.714 | 20,911.714 | 20,911.714 |
| | SBIC | 4893.099 | 4893.099 | 4893.099 | 21,658.855 | 21,658.855 | 21,658.855 |

**Table 2.** The properties of the CLS, FGLS, and 2SQML estimators for finite-lag specifications, when data is generated according to INARFIMA $(0, d, 0)$ models with $d = 0.25$ and lag (m) = 70.

| Lag | Parameters | T = 2000 and d = 0.25 | | | T = 10,000 and d = 0.25 | | |
|---|---|---|---|---|---|---|---|
| | | CLS | FGLS | ML | CLS | FGLS | ML |
| M10 | $\hat{d}$ | 0.397 | 0.397 | 0.397 | 0.404 | 0.404 | 0.404 |
| | (s.e.) | 0.001 | 0.004 | 0.005 | 0.000 | 0.002 | 0.002 |
| | BIAS | 0.147 | 0.147 | 0.147 | 0.154 | 0.154 | 0.154 |
| | MSE | 17.428 | 17.428 | 17.428 | 17.469 | 17.469 | 17.469 |
| | $Q_{LB100}$ | 252.360 | 252.360 | 252.360 | 861.417 | 861.418 | 861.418 |
| | $Q_{LB200}$ | 330.591 | 330.592 | 330.591 | 951.603 | 951.604 | 951.603 |
| | AIC | 5705.874 | 5705.875 | 5705.874 | 28,504.208 | 28,504.210 | 28,504.209 |
| | SBIC | 5778.484 | 5778.485 | 5778.484 | 28,594.522 | 28,594.523 | 28,594.523 |
| M30 | $\hat{d}$ | 0.292 | 0.292 | 0.292 | 0.298 | 0.298 | 0.298 |
| | (s.e.) | 0.001 | 0.003 | 0.003 | 0.000 | 0.001 | 0.001 |
| | BIAS | 0.042 | 0.042 | 0.042 | 0.048 | 0.048 | 0.048 |
| | MSE | 16.219 | 16.219 | 16.219 | 16.202 | 16.202 | 16.202 |
| | $Q_{LB100}$ | 173.555 | 173.555 | 173.555 | 434.016 | 434.016 | 434.016 |
| | $Q_{LB200}$ | 246.801 | 246.801 | 246.801 | 511.413 | 511.413 | 511.413 |
| | AIC | 5599.729 | 5599.729 | 5599.729 | 27,861.795 | 27,861.795 | 27,861.795 |
| | SBIC | 5804.357 | 5804.357 | 5804.357 | 28,116.315 | 28,116.316 | 28,116.315 |
| M50 | $\hat{d}$ | 0.260 | 0.260 | 0.260 | 0.264 | 0.264 | 0.264 |
| | (s.e.) | 0.001 | 0.003 | 0.003 | 0.000 | 0.001 | 0.001 |
| | BIAS | 0.010 | 0.010 | 0.010 | 0.014 | 0.014 | 0.014 |
| | MSE | 15.951 | 15.951 | 15.951 | 15.956 | 15.956 | 15.956 |
| | $Q_{LB100}$ | 152.446 | 152.446 | 152.446 | 343.686 | 343.686 | 343.686 |
| | $Q_{LB200}$ | 222.466 | 222.466 | 222.466 | 418.824 | 418.824 | 418.824 |
| | AIC | 5588.694 | 5588.694 | 5588.694 | 27,738.271 | 27,738.271 | 27,738.271 |
| | SBIC | 5925.340 | 5925.340 | 5925.340 | 28,156.998 | 28,156.998 | 28,156.998 |
| M70 | $\hat{d}$ | 0.242 | 0.242 | 0.242 | 0.245 | 0.245 | 0.245 |
| | (s.e.) | 0.001 | 0.003 | 0.003 | 0.000 | 0.001 | 0.001 |
| | BIAS | −0.008 | −0.008 | −0.008 | −0.005 | −0.005 | −0.005 |
| | MSE | 15.845 | 15.845 | 15.845 | 15.848 | 15.848 | 15.848 |
| | $Q_{LB100}$ | 136.373 | 136.373 | 136.373 | 294.513 | 294.513 | 294.513 |
| | $Q_{LB200}$ | 212.107 | 212.107 | 212.107 | 370.136 | 370.136 | 370.136 |
| | AIC | 5596.148 | 5596.148 | 5596.148 | 27,694.288 | 27,694.288 | 27,694.288 |
| | SBIC | 6064.812 | 6064.812 | 6064.812 | 28,277.222 | 28,277.222 | 28,277.223 |
| M90 | $\hat{d}$ | 0.230 | 0.230 | 0.230 | 0.233 | 0.233 | 0.233 |
| | (s.e.) | 0.001 | 0.003 | 0.003 | 0.000 | 0.001 | 0.001 |
| | BIAS | −0.020 | −0.020 | −0.020 | −0.017 | −0.017 | −0.017 |
| | MSE | 15.909 | 15.909 | 15.909 | 15.809 | 15.809 | 15.809 |
| | $Q_{LB100}$ | 132.433 | 132.433 | 132.433 | 268.457 | 268.457 | 268.457 |
| | $Q_{LB200}$ | 210.826 | 210.826 | 210.826 | 343.640 | 343.640 | 343.640 |
| | AIC | 5624.678 | 5624.678 | 5624.678 | 27,691.884 | 27,691.884 | 27,691.884 |
| | SBIC | 6225.360 | 6225.360 | 6225.360 | 28,439.025 | 28,439.025 | 28,439.025 |

**Table 3.** The properties of the CLS, FGLS, and 2SQML estimators for finite-lag specifications, when data is generated according to INARFIMA $(0, d, 0)$ models with $d = 0.4$ and lag (m) = 70.

| Lag | Parameters | T = 2000 and d = 0.4 | | | T = 10,000 and d = 0.4 | | |
|---|---|---|---|---|---|---|---|
| | | CLS | FGLS | ML | CLS | FGLS | ML |
| M10 | $\hat{d}$ | 0.598 | 0.598 | 0.598 | 0.605 | 0.605 | 0.605 |
| | (s.e.) | 0.001 | 0.004 | 0.005 | 0.000 | 0.002 | 0.002 |
| | BIAS | 0.198 | 0.198 | 0.198 | 0.205 | 0.205 | 0.205 |
| | MSE | 41.798 | 41.798 | 41.798 | 40.197 | 40.197 | 40.197 |
| | $Q_{LB100}$ | 549.235 | 549.235 | 549.235 | 1949.879 | 1949.878 | 1949.877 |
| | $Q_{LB200}$ | 665.178 | 665.177 | 665.177 | 2136.378 | 2136.377 | 2136.375 |
| | AIC | 7349.147 | 7349.147 | 7349.146 | 36,337.152 | 36,337.151 | 36,337.149 |
| | SBIC | 7421.757 | 7421.757 | 7421.756 | 36,427.466 | 36,427.465 | 36,427.463 |

<div align="center">**Table 3.** *Cont.*</div>

| Lag | Parameters | *T* = 2000 and *d* = 0.4 | | | *T* = 10,000 and *d* = 0.4 | | |
|---|---|---|---|---|---|---|---|
| | | CLS | FGLS | ML | CLS | FGLS | ML |
| M30 | $\hat{d}$ | 0.461 | 0.461 | 0.461 | 0.463 | 0.463 | 33.105 |
| | (s.e.) | 0.001 | 0.004 | 0.004 | 0.000 | 0.002 | 0.331 |
| | BIAS | 0.061 | 0.061 | 0.061 | 0.063 | 0.063 | 0.063 |
| | MSE | 34.030 | 34.030 | 34.030 | 33.105 | 33.105 | 33.105 |
| | $Q_{LB100}$ | 340.270 | 340.270 | 340.270 | 964.972 | 964.973 | 964.972 |
| | $Q_{LB200}$ | 436.152 | 436.152 | 436.152 | 1108.871 | 1108.872 | 1108.871 |
| | AIC | 7064.123 | 7064.124 | 7064.123 | 34,917.597 | 34,917.598 | 34,917.597 |
| | SBIC | 7268.751 | 7268.752 | 7268.751 | 35,172.117 | 35,172.119 | 35,172.117 |
| M50 | $\hat{d}$ | 0.410 | 0.410 | 0.410 | 0.412 | 0.412 | 0.412 |
| | (s.e.) | 0.001 | 0.004 | 0.004 | 0.000 | 0.002 | 0.002 |
| | BIAS | 0.010 | 0.010 | 0.010 | 0.012 | 0.012 | 0.012 |
| | MSE | 32.758 | 32.758 | 32.758 | 31.692 | 31.692 | 31.692 |
| | $Q_{LB100}$ | 301.374 | 301.375 | 301.374 | 742.921 | 742.922 | 742.921 |
| | $Q_{LB200}$ | 393.018 | 393.018 | 393.018 | 878.651 | 878.652 | 878.651 |
| | AIC | 7017.857 | 7017.858 | 7017.857 | 34,553.008 | 34,553.010 | 34,553.008 |
| | SBIC | 7354.503 | 7354.504 | 7354.503 | 34,971.736 | 34,971.737 | 34,971.736 |
| M70 | $\hat{d}$ | 0.382 | 0.382 | 0.382 | 0.384 | 0.384 | 0.384 |
| | (s.e.) | 0.001 | 0.004 | 0.003 | 0.000 | 0.002 | 0.002 |
| | BIAS | −0.018 | −0.018 | −0.018 | −0.016 | −0.016 | −0.016 |
| | MSE | 31.907 | 31.907 | 31.907 | 31.152 | 31.152 | 31.152 |
| | $Q_{LB100}$ | 275.290 | 275.291 | 275.290 | 657.154 | 657.154 | 657.154 |
| | $Q_{LB200}$ | 365.202 | 365.202 | 365.202 | 789.378 | 789.379 | 789.378 |
| | AIC | 6988.464 | 6988.464 | 6988.464 | 34,418.475 | 34,418.476 | 34,418.475 |
| | SBIC | 7457.128 | 7457.128 | 7457.128 | 35,001.409 | 35,001.410 | 35,001.409 |
| M90 | $\hat{d}$ | 0.364 | 0.364 | 0.364 | 0.364 | 0.364 | 0.364 |
| | (s.e.) | 0.001 | 0.003 | 0.004 | 0.000 | 0.002 | 0.001 |
| | BIAS | −0.036 | −0.036 | −0.036 | −0.036 | −0.036 | −0.036 |
| | MSE | 31.430 | 31.430 | 31.430 | 30.826 | 30.826 | 30.826 |
| | $Q_{LB100}$ | 255.701 | 255.701 | 255.701 | 602.208 | 602.208 | 602.208 |
| | $Q_{LB200}$ | 346.677 | 346.677 | 346.677 | 733.237 | 733.238 | 733.237 |
| | AIC | 6980.709 | 6980.709 | 6980.709 | 34,340.862 | 34,340.862 | 34,340.862 |
| | SBIC | 7581.391 | 7581.391 | 7581.391 | 35,088.002 | 35,088.003 | 35,088.002 |

## 5. Data and Descriptive

In this paper, we use the same data set used by Quoreshi (2014). The reason for this is that we introduce Quasi-Maximum Likelihood Method, which has not been considered in the study and the studies on count data earlier due to unknown underlying distribution. We intend to employ this method on the same data set and replicate the previous study to compare with results emerged from employing this method. Quoreshi (2014) has downloaded the tick-by-tick data for Ericsson B and AstraZeneca from the Ecovision system and the data are later filtered to generate transaction data which are counts. The stocks are frequently traded and have the highest turnovers at the Stockholmsbörsen. The two stock series are collected for the period 5 November–12 December 2002. Due to a technical problem in downloading data there are no data for 12 November in the time series and the first captured minutes of 5 December is 1037. Since we are interested in capturing the number of ordinary transactions, we have deleted all trading before 0935 (trading opens at 0930) and after 1714 (order book closes at 1720). The transactions in the first few minutes are subject to a different trading mechanism while there is practically no trading after 1714. The data are aggregated into one minute intervals of time. For high frequency data, researchers usually use one, two, five, or ten minute intervals of time and the choice is rather arbitrary. There are altogether 11,960 observations for both the Ericsson B and AstraZeneca series. The series together with their autocorrelation and partial-autocorrelation functions and histograms are exhibited in Figure 1. There are frequent zero frequencies in both series, especially in the AstraZeneca series, and hence the application of count data modeling is called for. The counts in both series fluctuate around their means which is an indication of mean reverting processes. The autocorrelation functions

for both series suggest fractional integration which implies long memory. The histograms exhibit the distribution of counts and the possible empirical densities of the counts. Even with relatively large sample size, the distributions appear far from being normally distributed. On the other hand, the distributions appear to be similar to Poisson distribution; both variances are greater than their respective means (Figure 1). Hence, there is no scope for employing known distributions.

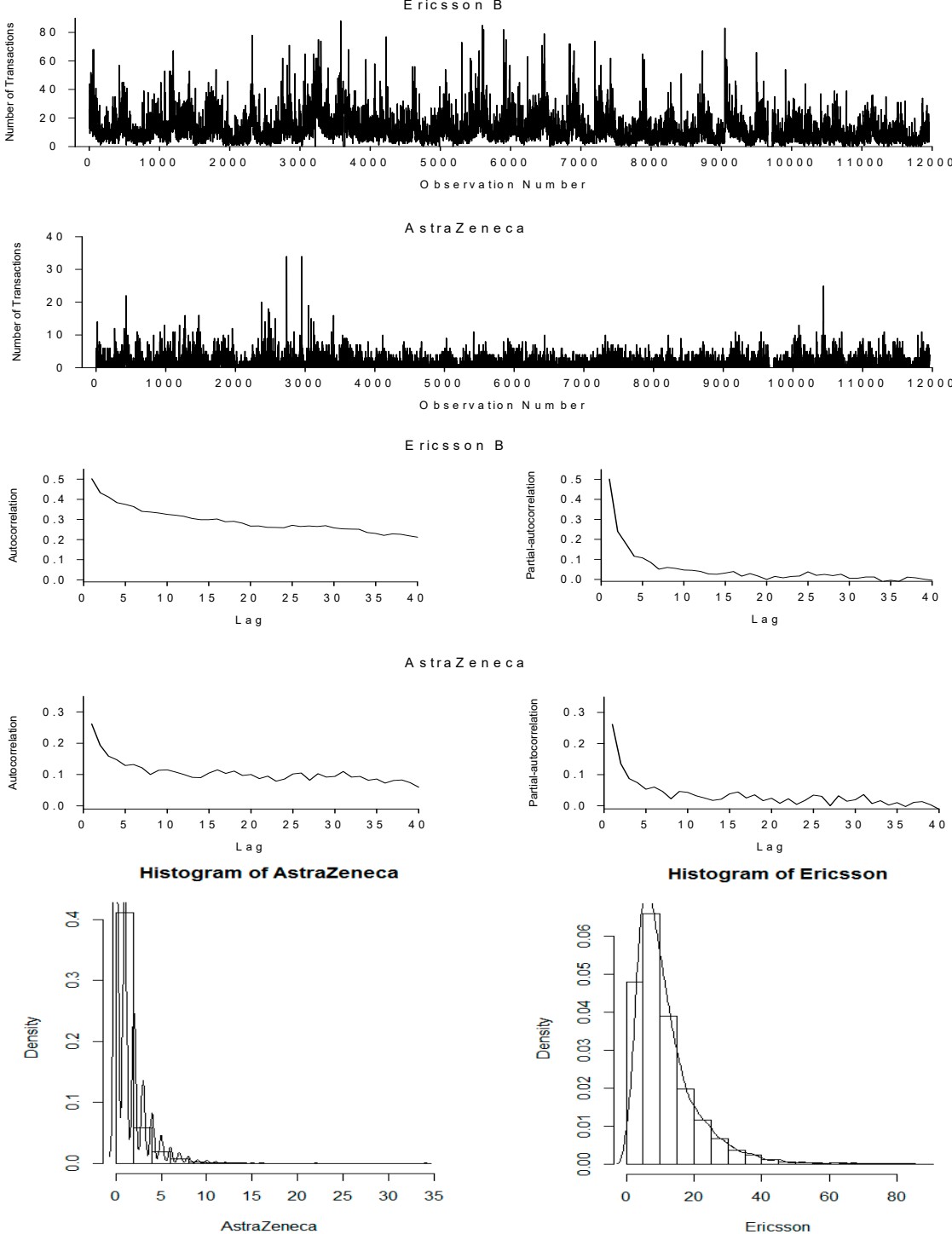

**Figure 1.** The time series of Ericsson B (mean 11.73 and variance 84.86) and AstraZeneca (mean 1.33 and variance 3.75) and their autocorrelation and partial-autocorrelation functions.

## 6. Empirical Results

CLS, FGLS, and 2SQML are employed for estimation with lag length of 70 following the suggestions of Brännäs and Quoreshi (2010). It is to be noted that the AIC and SBIC criteria are not applicable in the context of long memory (Brännäs and Quoreshi 2010; Quoreshi 2014), which is also supported by the Monte Carlo Experiment. The results of the empirical studies are presented in Table 4. Empirically, we find evidence for long memory property ($\hat{d} < 0.5$) for both Ericsson B and AstraZeneca series (see Table 4). The series for both AstraZeneca and Ericsson has mean reversion property and is covariance stationary. The findings suggest that the impact of macroeconomic news or rumors in one point of time has a persistence impact on future transactions. It may be recalled that news disseminated through formal channels which may have impact on overall stock markets and specifically on a particular stock is termed macroeconomic news, such as news on interest rates or unemployment statistics for a country which may influence all stocks. Rumors are information that spread through unofficial channels yet is related to macroeconomic news or a particular stock. For AstraZeneca, we find that CLS perform best while 2SQML performs better than FGLS in terms of eliminating serial correlation. Considering the standard error of the parameters ($\hat{\lambda}$, $\hat{d}$), we find that 2SQML performs best among the three estimators, while CLS performs better than FGLS. For Ericsson B, we find that 2SQML performs best in terms of both eliminating serial correlations and minimum standard error for the parameters. All other corresponding estimates turn out equal up to 3 decimals estimated by the three estimators.

**Table 4.** CLS, FGLS, and 2SQML estimates for Ericsson and AstraZeneca.

|  | Ericsson | | | AstraZeneca | | |
|---|---|---|---|---|---|---|
|  | **CLS** | **FGLS** | **ML** | **CLS** | **FGLS** | **ML** |
| $\hat{d}$ | 0.324 | 0.324 | 0.324 | 0.204 | 0.204 | 0.204 |
| (s.e) | 0.008 | 0.008 | 0.008 | 0.012 | 0.012 | 0.011 |
| $\hat{\lambda}$ | 2.625 | 2.625 | 2.625 | 0.507 | 0.507 | 0.507 |
| (s.e) | 0.091 | 0.092 | 0.088 | 0.027 | 0.027 | 0.026 |
| Var | - | - | 54.722 | - | - | 3.303 |
| (s.e.) | - | - | 1.809 | - | - | 0.151 |
| AIC | 48,010.160 | 48,010.160 | 48,010.160 | 14,433.565 | 14,433.565 | 14,433.565 |
| SBIC | 48,534.802 | 48,534.802 | 48,534.802 | 14,958.207 | 14,958.207 | 14,958.207 |
| $Q_{LB100}$ | 335.111 | 335.111 | 335.111 | 235.586 | 235.586 | 235.587 |
| $Q_{LB200}$ | 422.191 | 422.191 | 422.191 | 352.309 | 352.309 | 352.309 |
| MSE | 54.722 | 54.722 | 54.722 | 3.303 | 3.303 | 3.303 |

## 7. Concluding Remarks

This paper introduces Quasi-Maximum Likelihood Estimation of Integer-Valued Long Memory Model for unknown underlying distribution and the estimation procedures for QML have been discussed. The paper compares the 2SQML estimator with FGLS and CLS. In a Monte Carlo experiment, it is found that the 2SQML, FGLS, and CLS estimators perform equally well in terms of eliminating serial correlation. The empirical study suggests that CLS performs best for AstraZeneca, while 2SQML performs best for Ericsson B in terms of eliminating serial correlations. However, the 2SQML estimator performs better than both the CLS and FGLS in terms of minimum standard error for estimates of the parameters for both Ericsson B and AstraZeneca, although CLS perform best followed by 2SQML in the simulation study. Note that the data in the simulation study is equidispersed since the data is generated from the Poisson distribution, while the data for the empirical study is overdispersed. The results of the study collectively may indicate that 2SQML estimator is relatively more efficient compared to CLS and FGLS for overdispersed data. The empirical results suggest that both series have long memory properties, which implies that the impact of macroeconomic news or rumors in one point of time has a persistence impact on future transactions.

**Author Contributions:** R.U. Contributed to literature review, estimation, simulation and data Analysis. N.M.K. contributed to estimation, simulation and commenting on overall paper. A.M.M.S.Q. designed, supervised and wrote the paper.

**Funding:** This research received no external funding.

**Conflicts of Interest:** The authors declare no conflict of interest.

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
