# Peer review of "Quasi-Maximum Likelihood Estimation for Long Memory Stock Transaction Data—Under Conditional Heteroskedasticity Framework"

_jrfm, doi:10.3390/jrfm12020074_

Round 1
Reviewer 1 Report
The paper compares the 2SQML estimator with FGLS and CLS. 2SQML estimator performs better than both CLS and FGLS in terms of minimum standard error for estimates of the parameters in two stock series, although CLS performs best followed by 2SQML in the simulation study.
The paper is well written and structured. However, the very first thing that one notices is that the paper clearly lacks more depth as for its motivation, which is quite poor. Besides, the discussion of the literature should also be enriched.
Why analyze high frequency data for these two stock series (November 5, 2002 to December 12, 2002)? What is the novel contribution that the paper seeks to have about our knowledge regarding long-memory phenomenon in the context of stock market? What is the economic/financial interpretation of their results? And what exactly has the paper contributed to our knowledge of the relevance of financial time series properties?
Author Response
Dear Reviewer,
I have attached a file.

Reviewer 2 Report
The authors are familiar with the literature and the empirical work is solid. However, the overall feeling is that the paper is done in a rush. The following is my specific comments that aim to improve the paper. The authors want to make it very clear about
(1) What is the criteria for assessing "goodness" of an estimator? Are there many criteria or just one? If there are several criteria, which one the authors are focusing on? What is the standard criteria for "goodness"? Eliminating serial correlation in the residuals? The paper should also mention a paper by Bollerslev et al. that raw returns scaled by realized volatility conforms to a normal distribution. What is the advantage of parameterized models ?
(2) What other studies have shown? What is the criteria and what are the estimators?
Although people in this area are familiar with these issues, people not in this area need some
time to find out and catch up with the literature. Therefore the authors need to make it very
clear. Right now the paper is not well packaged. It is very confusing . Basically the paper
assumes readers are familiar with the literature.
(3) Why focus on the sample period of an old study by Quorershi (2014)? Why not update the
data until 20181231? The author mentioned "The stock series collected represent the period the period between November 5, 2002 to December 2002." But later ,the authors mention "data for November 2012 is not available". What is the correct sample period? The authors need to be careful on these small issues.
(4) Why focus on two stocks only? Why not use the tick data for a broad market index such as SP 500. SP500 minute by minute data are available on WRDS and can be easily downloaded. Addressing SP500 tick data will be much more important than addressing two stocks in Sweden.
(5) All tables should line up decimal positions. There is no need for shared rows in Table 4 and others. 3-decimal positions are good enough. Additional ones are not necessary.
(6) The authors should provide some paragraphs explaining what is going on in Tables 1, 2, and 3. It is hard to immediately see what is being done in TAbles 1,2, and 3.
Author Response
I have attached a file.

Round 2
Reviewer 2 Report
Please make sure the article conforms to journal style, including tables, section titles.